# Biologic Functions of Hydroxychloroquine in Disease: From COVID-19 to Cancer

**DOI:** 10.3390/pharmaceutics14122551

**Published:** 2022-11-22

**Authors:** Britney Niemann, Amanda Puleo, Conley Stout, Justin Markel, Brian A. Boone

**Affiliations:** 1Department of Surgery, West Virginia University, Morgantown, WV 26506, USA; 2Department of Microbiology, Immunology and Cell Biology, West Virginia University, Morgantown, WV 26506, USA

**Keywords:** Hydroxychloroquine, Chloroquine, malaria, cancer, autophagy, autoimmune

## Abstract

Chloroquine (CQ) and Hydroxychloroquine (HCQ), initially utilized in the treatment of malaria, have now developed a long list of applications. Despite their clinical relevance, their mechanisms of action are not clearly defined. Major pathways by which these agents are proposed to function include alkalinization of lysosomes and endosomes, downregulation of C-X-C chemokine receptor type 4 (CXCR4) expression, high-mobility group box 1 protein (HMGB1) inhibition, alteration of intracellular calcium, and prevention of thrombus formation. However, there is conflicting data present in the literature. This is likely the result of the complex overlapping pathways between these mechanisms of action that have not previously been highlighted. In fact, prior research has focused on very specific portions of particular pathways without describing these in the context of the extensive CQ/HCQ literature. This review summarizes the detailed data regarding CQ/HCQ’s mechanisms of action while also providing insight into the overarching themes. Furthermore, this review provides clinical context to the application of these diverse drugs including their role in malaria, autoimmune disorders, cardiovascular disease, thrombus formation, malignancies, and viral infections.

## 1. Introduction

Chloroquine (CQ) and its derivative, hydroxychloroquine (HCQ), are well-known, multi-use drugs with applications in anti-malarial and anti-viral treatment, autoimmune diseases, and neoplastic processes [1,2,3,4]. While CQ and HCQ have been used for decades, new uses continue to be discovered. However, despite their widespread use, the mechanism of action is poorly understood. In fact, numerous mechanisms with overlapping pathways have been proposed. This review provides a comprehensive summary of these mechanisms in order to enhance understanding of CQ/HCQ, allowing for more directed research in its clinical applications. Additionally, we will review the clinical data available for these drugs in a multitude of disease processes. 

### 1.1. History of Chloroquine/Hydroxychloroquine

Initially discovered and disregarded by the Germans secondary to toxicity, CQs effectiveness as an anti-malarial was independently discovered by a coalition of Allied private, military and federal researchers [5]. Its development demand came when previous sources of quinine were cut off after the attack on Pearl Harbor and following known documentation of CQ effectiveness in Vivax Malaria by the Army Medical Corps in 1946 [6]. By the 1960s, numerous accounts of resistant plasmodium falciparum were being reported [7,8,9]. Furthermore, as U.S. military involvement in Asia waned near the end of the 1960s, so was the government’s commitment to developing further anti-malarial research, and chloroquine use seemed to be coming to an end [8].

### 1.2. Repurposing

Although CQ’s effectiveness as an anti-malarial was deteriorating due to spreading resistance, alternative applications began to reveal themselves. In the 1950s, multiple reports were made documenting symptomatic relief in patients with Discoid Lupus Erythematosus after receiving CQ treatment for malaria [1,2]. Similarly, CQ and its derivative HCQ have long been utilized as a therapeutic agent in the treatment of rheumatoid arthritis (RA) [3,10]. In addition to autoimmune diseases, CQ/HCQ use has been expounded for viral illnesses as well. It was noted that CQ had a synergistic effect with common anti-retroviral medications for human immunodeficiency virus (HIV). An increase in CD-4 counts and p24 inhibition was observed when CQ was used in conjunction with anti-HIV medications [11,12]. Other viral illnesses have also been studied using CQ as a treatment for influenza, filoviruses, and both severe acute respiratory syndrome (SARS) and middle eastern respiratory syndrome (MERS) variants of coronaviruses [13,14,15,16,17]. Most recently, HCQ briefly gained marked prominence as a potential treatment for SARS CoV-2 (COVID) [17,18,19,20]. In addition, CQ/HCQ are now also being studied for their role in the treatment of a wide variety of cancers [4]. 

Chloroquine and hydroxychloroquine are orally bioavailable and have established safety profiles, which has led to substantial research into new applications. This nearly century-old drug has an additional benefit of interest to researchers and clinicians, in that it is extremely affordable [21]. 

## 2. Biochemistry and Pharmacology

Chloroquine and hydroxychloroquine are 4-aminoquinolines with anti-malarial, anti-viral, anti-inflammatory, and anti-cancer applications [22]. Structurally, they differ by only one hydroxyl group (Figure 1). Though CQ has been in use for the better part of a century, its pharmacokinetics were not studied in detail until approximately the 1980s. Using liquid chromatographic techniques with diethyl ether extraction to identify CQ and desethylchloroquine (main metabolite), Gustafsson et al. studied chloroquine concentrations in healthy individuals after single dose administration with either intravenous (IV) or per os (PO) formulation. They found the drug could be detected in urine samples 23–52 days after administration with massive volumes of distribution (Vd) ranging from 111–285 L/kg [23]. Large Vd were further validated by Frisk-Holmberg et al. who showed up to 800 L/kg when calculated by plasma concentrations; however, whole blood concentrations were 8–10 times higher and, consequently, had Vd approximately 10 times lower [24]. This effect is explained by the finding that chloroquine is concentrated in erythrocytes and is approximately 2–5 times higher in red blood cells than in plasma [23]. Further explanation of such large Vd is likely due to the basic nature of chloroquine and its affinity for lysosomal uptake [25,26]. Bioavailability of solution and pill form has been reported ranging from 78% to nearly 100%, with the higher value coming from later studies. The high bioavailability is attributed to rapid distribution into erythrocytes and thus low plasma levels exposed to first pass hepatic metabolism [23,27]. Similar findings were observed in studies of chloroquine malaria treatment in children [28]. Pregnancy has been shown to decrease half-lives and Vd [29]. The limited amount of chloroquine and desethylchloroquine in plasma is bound to albumin [30,31]. Furthermore, plasma levels varied by enantiomer with (S)-chloroquine plasma levels being approximately 66% bound and (R)-chloroquine 42% bound [32]. Unmetabolized CQ is excreted primarily in urine [33]. A detailed reviewed of chloroquine pharmacokinetics was previously completed by White and more recently by Ducharme and Farinotti [33,34].

### 2.1. Side Effects

Numerous sides effects have been reported with substantial increase in prevalence when the dose was 10–15 mg/kg vs. 2–5 mg/kg [35,36]. One common side effect is retinopathy. Risk factors for retinopathy with HCQ include daily dose, duration of use, concurrent tamoxifen use, and the presence of kidney disease [37,38]. The risk of retinopathy is difficult to quantify with CQ, as blood levels related to risk are not always commensurate to blood levels necessary for treatment efficacy. Some have advocated for dosing based on actual weight, rather than ideal, due to the risk of overdosing thin patients who suffer from chronic inflammatory conditions and tend to be underweight [39]. Current ophthalmological recommendations for screening are the completion of a baseline fundoscopic exam, with annual exams beginning after 5 years of use [40]. Reversible neuromyotoxicity with characteristic vacuolar myopathy on electron microscopy has also been well-reported [41,42]. Other side effects include blurred vision (54%), pruritus (22%), paresthesia (6%), and insomnia (46%) [42,43]. Pruritus, in particular, has been documented and is likely dependent on serum level rather than total dose administered [44,45,46]. Neuropsychiatric effects have ranged from insomnia to psychosis, possibly as a result of acetylcholinesterase inhibition [47,48]. Lastly, QT prolongation and torsade de pointes have been reported with increased incidence in patients with heart failure, atrial fibrillation, and chronic kidney disease. Therefore, cardiac evaluation should be considered prior to initiation [49,50].

### 2.2. Contraindications

The only contraindications for drug administration are known hypersensitivity to 4-aminoquinoline compounds or preexisting retinal disease [51]. 

## 3. Mechanisms of Action

Researchers have investigated the mechanism of action of CQ and HCQ. However, years of exploration have revealed multiple complex and diverse mechanisms by which these drugs function. Overall, their biologic functions can be broken down into 5 major categories, depicted throughout Figure 2, Figure 3, Figure 4, Figure 5 and Figure 6: alkalinization of lysosomes and endosomes; downregulation of C-X-C chemokine receptor type 4 (CXCR4) expression; high-mobility group box 1 protein (HMGB1) inhibition; alteration of intracellular calcium; and prevention of thrombus formation (Figure 2).

### 3.1. Alkalinization of Lysosomes and Endosomes

CQ and HCQ’s basic properties allow for the drugs to accumulate in, and alkalinize, the acidic environments of lysosomes and endosomes (Figure 3). Each of these organelles contribute to the processes of cell death and cell signaling. Endosomes play an important role in the entry and replication of several viruses, thrombus formation in autoimmune disorders, and cell signaling through endocytic Toll-like receptors (TLRs). Lysosomes, on the other hand, drive autophagy and cell death.

Autophagy is an important cellular process where catabolism of cellular components occurs in the settings of nutrient deprivation, hypoxia, and other cell stressors including chemotherapy and radiation [52]. This process can serve multiple purposes, such as energy production in hypoxic and nutrient-deprived environments, and clearance of damaged organelles and reactive oxygen species (ROS) [53].

Based on the initial stressor signal, different pathways are involved in activating autophagy. The common pathway converges on the formation of the autophagosome, a double membrane structure that encloses cellular components. The autophagosome then fuses with a lysosome, allowing for degradation of its contents via lysosome hydrolases [53,54]. CQ/HCQs alkalinization of lysosomes prevents this fusion, as well as impairs the function of lysosomal hydrolases, resulting in autophagy inhibition and impaired lysosome hydrolase function (Figure 3) [55]. CQ’s known risk of ocular toxicity has been attributed to this dysfunction of lysosome hydrolases in retinal pigment epithelium (RPE). Lysosomal dysfunction has been shown to lead to an accumulation of lipofuscin, which can lead to retinal toxicity [56,57].

The consequences of autophagy inhibition are numerous. Autophagy has been implicated in carcinogenesis, disease progression, and even metastasis [55,58,59,60]. Tumors with high proliferation rates often outgrow their blood supply and thus, their source of nutrients. Autophagy can serve as a source of fuel in these settings, and therefore, it is no surprise increased rates of autophagy are found in many cancers [55,58,59,60,61]. Pancreatic cancer, for example, has a strong link to increased autophagy and tumor grade, resulting in poor prognosis [58,62]. Yang et al. demonstrated inhibition of autophagy, through genetic means or use of CQ, led to accumulation of ROS which induced DNA damage and decreased cancer cell growth in vitro. Furthermore, inhibition of autophagy with CQ resulted in tumor regression and prolonged survival in mouse models [63]. Likewise, cancer stem cells, which play an important role in tumor initiation, metastasis, recurrence, and chemoresistance, utilize autophagy, with CQ treatment leading to regression and improved outcomes [64,65].

Autophagy may also mediate resistance to chemotherapy. Upregulation of autophagy is seen following multiple chemotherapy agents [66]. To clarify the role of autophagy in chemotherapy resistance, genetic silencing of autophagy-related genes (ATGs) has been tested in the setting of chemoresistant cancer cell lines. Genetic silencing was shown to sensitize previously chemoresistant cells to therapy [67]. Hashimoto et al. demonstrated an increase in autophagy following treatment of pancreatic cancer cells with either 5-fluorouracil (5-FU) or gemcitabine. However, treatment in combination with chloroquine reduced autophagy and potentiated the antiproliferative effects of 5-FU and gemcitabine [68]. Glioblastoma cells were also found to utilize higher rates of autophagy to overcome treatment with Bevacizumab, a monoclonal antibody to vascular endothelial growth factor (VEGF). Chloroquine and hydroxychloroquine reverse this resistance in multiple studies [69,70]. These studies indicate that although CQ/HCQ alone have shown antitumor effects, they may be best utilized as a combination therapy.

In addition to the effect on oncologic cells from autophagy, it also appears to play an important role in activating cancer supporting cells. Pancreatic stellate cells (PSCs) protect the tumor from the immune system’s antitumor defense by creating a strong, fibrous stroma around the tumor which decreases T cell infiltration. Endo et al. linked autophagy with PSC activation and, as with pancreatic cancer cells, associated this increased rate of autophagy to a poor prognosis. By inhibiting autophagy with chloroquine, the authors were able to demonstrate conversion of PSCs to a quiescent state as well as a decrease in extracellular matrix accumulation and tumor volumes [71]. Similarly, cancer associated fibroblasts (CAFs) in breast cancer enhance the growth and metastatic potential of breast tumors. Caveolin-1 (Cav-1) is a structural protein and transformation suppressor expressed by healthy fibroblasts. Breast cancer cells are able to downregulate the expression of Cav-1, leading to early disease progression and poor prognosis. Interestingly, CQ was able to restore Cav-1 expression, indicating cancer cells may use autophagy to degrade antitumor structures [61].

Autophagy also plays a role in multiple autoimmune diseases via antigen processing and presentation, T cell activation, and cytokine processing [72,73,74]. Overactivation of T cells results in the body incorrectly targeting self-antigens leading to cell death, extensive inflammation, and organ damage [75]. Through autophagy inhibition, CQ and HCQ prevent autoantigen presentation in antigen presenting cells and B cells, resulting in decreased T cell activation [74]. Rheumatoid arthritis (RA) results in a dysregulation in autophagy and is characterized by synovial inflammation, increased bone catabolism, and damage to cartilage and bone. RA patients develop autoantibodies, often to citrullinated proteins. Fibroblast-like synoviocytes (FLS) are found infiltrating cartilage and bone surfaces and can deposit collagen and α-smooth muscle actin causing synovial fibrosis. High levels of autophagy in RA FLS allow for prolonged survival and correlate with increased levels of antibodies against citrullinated proteins [72]. 

While lysosomes can support cell survival through autophagy, they can also promote cell death. Cell death can occur through cell necrosis or apoptosis, both of which can be impacted by lysosomes. Lysosomal death is initiated by lysosomal membrane permeabilization (LMP) which allows the translocation of lysosomal enzymes into the cytoplasm instigating cell death [76]. CQ and HCQ are capable of causing permeabilization of not only lysosomal membranes, but also mitochondrial and plasma membranes (Figure 3). Boya et al. showed that HCQ accumulation in lysosomes resulted in increased lysosomal volume followed by lysosomal, mitochondrial, and plasma membrane permeabilization [77]. Extensive permeabilization, as well as lysosomal hydrolase activity within the cytoplasm, resulted in cell death. LMP induction may be an additional mechanism by which CQ overcomes resistance to chemotherapy. For example, patients with non-small-cell lung cancer (NSCLC) unable to receive immunotherapy often receive chemotherapy. However, resistance forms quickly. CQ was shown to induce LMP leading to apoptosis of NSCLC cells [78]. This effect has also been demonstrated with CQ treatment in conjunction with PI3K/mTOR inhibitors [79,80].

Chloroquine’s role in the treatment of malaria has also been attributed to the alkalinization of a type of secondary lysosome called a digestive vesicle (DV) [81]. Malaria, caused by different species of the parasite Plasmodium, is characterized by parasitic invasion of host red blood cells (RBCs). Plasmodium degrades hemoglobin within DVs and utilizes the amino acid products. Heme is also released during this process and is toxic to parasites. However, in the acidic environment of DVs it is quickly converted to the nontoxic hemozoin. This process is inhibited in the setting of CQ induced alkalinization of DVs resulting in heme toxicity to parasites. Plasmodium quickly adapts, though, resulting in widespread CQ resistance. This is secondary to a mutation in the plasmodium falciparum chloroquine resistance transporter gene (pfcrt) which allows for the efflux of CQ out of DV through a transporter protein [82]. Specifically, the transporter takes on a configuration that produces an overall negative charge, attracting and sequestering positively charged compounds such as CQ [83].

Alkalinization of acidic compartments by CQ also impacts endosome function. Endosomes are a critical part of endocytosis, a process which propagates cell signaling and allows them to internalize aspects of the surrounding environment. Viruses commonly utilize endocytosis to gain entry into a cell. CQ has been shown to decrease intracellular viral accumulation of multiple viruses, including Borna virus, HIV, Hepatitis A, Zika virus, Hepatitis C, Dengue virus, and Ebola [11,16,84,85,86,87]. Viral replication is also dependent on organelle pH for intracellular trafficking, unpacking, and post-translational modification [86]. Importantly, other antiviral mechanisms have been proposed separate from the effects of CQ on organelle pH. CQ has been shown to inhibit glycosylation, a necessary process for the glycosylation of viral envelopes and subsequent release [11]. Another possible mechanism is proposed by the inhibition of arachidonic acid metabolism and activation of NFκB, thus decreasing transcription of viral DNA [88]. 

Some Toll-like receptors (TLRs) depend on endosomal function for the transmission of their signal. TLRs are transmembrane proteins with important functions in innate immunity and inflammation. The proteins are located either on the plasma membrane or endosomal membrane. Endocytic TLRs 3, 7, 8, and 9 require internalization of ligands to stimulate activity, a process which is inhibited by compartment alkalinization by CQ (Figure 3) [89,90,91,92,93,94]. In fact, Rutz et al. identified CQs ability to inhibit TLR9 signaling to be pH dependent, supporting this proposed mechanism [95]. Endocytic TLRs are involved in sepsis-induced mortality and acute kidney injury (AKI) in mouse models. Treatment with CQ decreased AKI, TLR protein in the spleen, and systemic inflammation as well as improved the survival rate [96]. Likewise, treatment with CQ prevented bacterial DNA-induced TLR signaling of the inflammatory response to sepsis [97]. TLR inhibition has many downstream effects, including reduced cytokine production, and impaired recognition of immune complexes by endosomal TLRs in autoimmune diseases [98]. Additionally, TLR9 may have a role in the pathogenesis of type I diabetes. CQ treatment decreased development of diabetes and improved islet cell function. Of note, there is evidence that CQs effect on TLRs may extend beyond pH modifications. Kuznik et al. recently demonstrated CQs effect on TLRs is present even with only minimal changes in endosomal pH. In fact, CQ could directly bind ligands in order to prevent their binding to TLRs. Further revealed was CQ’s capability to directly bind TLR ligands, preventing their binding to receptors. CQ was found to inhibit the function of TLRs 3, 7, and 9, but acted as an agonist for TLR8 [99]. Similarly, Zhang et al. demonstrated the reversibility of CQs effect via addition of a TLR9 agonist, again contradicting the theory of alkalinization as the sole source of the mechanism [100].

Finally, CQ has shown efficacy in inhibiting the formation of neutrophil extracellular traps (NETs), a phenomenon where neutrophils expel their intracellular contents as a mechanism to combat infection (Figure 4) [101]. NETs are composed of highly decondensed chromatin structures rich in histones, proteins and granular content [102]. Since their discovery, NETs have been implicated in many diseases, including autoimmune diseases, various cancers, and thrombus formation. NETosis is stimulated by a number of organisms and factors, including activated platelets, autoantibodies, IL-8, and cigarette smoke [101,102,103]. Therefore, it is no surprise that multiple pathways lead to NET formation. Autophagy, and TLRs 7 and 9—targets of CQ—are a few among the pathways. Additionally, activation of the NADPH oxidase (NOX) complex has been shown to contribute to NETosis. The NOX complex requires endosomal function, which may be another point at which CQ interferes with NETosis (Figure 3 and Figure 4) [102]. Lastly, the enzyme peptidyl arginine deiminase 4 (PAD4) plays an important role in NETosis and will be discussed further in a subsequent section (Figure 4) [102,104].

### 3.2. C-X-C Chemokine Receptor Type 4 

C-X-C chemokine receptor type 4 (CXCR4) is a chemokine receptor that, along with its ligand C-X-C motif chemokine ligand 12 (CXCL12), impacts many physiologic as well as pathologic processes. The CXCR4/CXCL12 axis is widely expressed throughout the human body, and their downstream effects of receptor binding result in gene transcription, cell proliferation and survival, and cellular adhesion and migration [105,106]. The embryonic vitality of the CXCR4 receptor and chemokine has been demonstrated in murine models; and the physiologic functions of embryogenesis, hematopoiesis, brain development, and leucocyte trafficking towards sites of inflammation have been well established [106,107,108,109,110]. Pathologically, a role in multiple autoimmune diseases, stroke, and the cellular entry of human immunodeficiency virus has also been studied [106,110,111,112]. In oncologic disease, CXCR4 has been found to be frequently overexpressed in malignant cells and linked to primary tumor growth, angiogenesis, tumor invasion of surrounding tissues, and metastasis [105,106,109,110,113,114,115]. Due to the pathologic function of this axis, it has gained attention from researchers searching for a viable inhibitor. Chloroquine-containing products have been found to downregulate CXCR4 expression (Figure 5) [105,116,117].

In 2012, Kim et al. observed decreased CXCR4-mediated pancreatic cancer cell signaling and proliferation in vitro [118]. Further in vitro experimentation by Balic et al. in 2014 showed a significant decrease in the number of circulating tumor cells in pancreatic cancer treated with chloroquine. The inhibition was found to reduce phosphorylation of extracellular signal-regulated kinase (ERK) and signal transducer and activator of transcription 3 (STAT3) and showed potential to assist with the control of metastatic disease (Figure 5) [119]. Inhibition of CXCR4 with CQ has also been shown to delay tumor progression in esophageal cancer in mice [120]. Through this pathway of inhibition, effects on tumor vasculature and immune system function have also been noted [121]. In 2016, Yu et al. published two studies where synthesized CQ was used to decrease cell surface expression of CXCR4 in oncologic cells and proved to have both antimetastatic properties in addition to causing less toxicity than its parent drug, hydroxychloroquine [122,123]. While this mechanism of action for chloroquine products has not been the most studied, there is evidence supporting its further research and how it may help the treatment of oncologic disease.

### 3.3. High-Mobility Group Box 1 Protein

High-mobility group box 1 protein is a DNA-binding protein with both intra- and extracellular functions through many receptors such as that found in advanced glycation end products (RAGE), T cell immunoglobulin domain and mucin domain-3, and TLR4. HMGB1s downstream effects are abundant and include the following: transcription regulation, autophagy initiation, carcinogenesis, angiogenesis, potentiation of inflammation and ischemia, cytokine production, hypercoagulability, NETosis, and sepsis [90,124,125,126,127,128,129,130,131]. In the presence of ROS, there is an upregulation of RAGE expression, which binds HMGB1 resulting in the activation of multiple pathways. First, TLR9 can be stimulated, resulting in the release of inflammatory cytokines (Figure 3). Second, IL-6 release activates STAT3, a process that can both enhance autophagy and increase CXCR4 expression. Third, the RAGE-HMGB1 complex also directly triggers autophagy (Figure 4). As discussed, CQ can impact multiple aspects of these pathways, including TLR9, CXCR4, and autophagy (Figure 2, Figure 3, Figure 4 and Figure 5). However, CQ has also been shown to inhibit release of HMGB1 in septic mice resulting in improved mortality (Figure 5) [128]. Furthermore, it prevented release of HMGB1 from monocytes following stimulation with LPS or IFNγ [132]. This impact has not yet been studied extensively in other pathologies but represents an additional potential target of CQ.

### 3.4. Alteration of Intracellular Calcium

Intracellular calcium stores are an important part of cell signaling with increased intracellular calcium levels resulting in signal propagation. Platelet aggregation, for instance, often requires alterations in intracellular calcium levels. Platelet aggregation can be induced by multiple stimulants, including phorbol-myristate-acetate (PMA), calcium (Ca) ionophores, and thrombin (Figure 6).

PMA and thrombin act via protein kinase C (PKC), resulting in a mobilization of intracellular calcium stores. On the other hand, Ca ionophores do not require membrane receptors and utilize influx of extracellular calcium. Ca ionophore and thrombin stimulation both induce release of phospholipase A2 (PLA2), leading to arachidonic acid (AA) liberation from plasma membrane phospholipids (Figure 6). The arachidonic acid cascade is critical for platelet aggregation as it yields thromboxane A2 (TXA2) which is important for aggregation and vasoconstriction. CQ is capable of inhibiting platelet aggregation secondary to PMA, Ca ionophore, and thrombin stimulation; however, it is less potent in relation to Ca ionophore stimulation. This difference may indicate that CQ plays more prominence on intracellular calcium than extracellular [133,134]. Research has shown CQ can also inhibit the arachidonic acid pathway via inhibition of PLA2, leading to reduction in TXA2 production [134,135].

PAD4 function has also been shown to be dependent on high intracellular calcium levels. PAD4 is an enzyme that citrullinates DNA histones resulting in decondensed chromatin. Its actions are essential to NETosis as PAD4 deficient mouse neutrophils are unable to form NETs [136]. PAD4 inhibitors also reduce NET formation in mouse and human neutrophils [137]. As discussed, NETs play an important role in multiple stages of cancer, autoimmune disease, and thrombus formation. PAD4 function disruption via alteration of intracellular calcium stores may be yet another mechanism by which HCQ inhibits NET formation (Figure 4).

### 3.5. Prevention of Thrombosis

Autoimmune diseases such as antiphospholipid syndrome (APS) and systemic lupus erythematosus (SLE) have increased rates of both arterial and venous thrombi [138]. In particular, APS is an autoimmune disorder characterized by recurrent thrombosis and pregnancy losses and can occur as a primary disorder or in conjunction with SLE. APS antibodies (aPL) include lupus anticoagulant, anticardiolipin antibodies, or anti-β2 glycoprotein I antibodies [139]. APS antibodies create a pro-thrombotic environment via multiple pathways, such as interference with annexin A5, an anticoagulant protein found in both adult vasculature and the placenta. The A5 protein forms a crystal that covers phospholipid membranes to prevent interaction with coagulation enzymes; however, aPLs bind annexin A5, thus inhibiting the formation of this protective shield. HCQ not only restores the original crystal layer, but also induces the formation of a second crystal layer over the anti-β2 glycoprotein I binding sites (Figure 4) [140]. Furthermore, HCQ is capable of preventing aPL binding to the phospholipid bilayer, as well as reversing the effects of aPLs, including platelet activation, increased TF expression, increased GPIIb/IIIa expression, and increased thrombin and thrombin receptor peptide agonist generation [141,142,143,144,145,146].

aPLs may also induce thrombus formation via activation of NADPH oxidase (NOX). NOX mediates multiple inflammatory pathways, including TNFα and IL-1β signaling, and has been shown to influence endothelial dysfunction following stimulation by aPL [143,147]. The NOX ligand-receptor complex requires entrance into the endosome in order for downstream signaling to occur resulting in reactive oxygen species (ROS) production and thrombus formation. As discussed, CQ and HCQ affect endosomal function (Figure 3); therefore, it is no surprise that HCQ is capable of inhibiting ROS and thrombi production [148,149]. Furthermore, HCQ reverses endothelial dysfunction secondary to aPL-induced endothelial nitric oxide synthase (eNOS) inhibition and upregulation of adhesion molecules (Figure 4) [148]. Endothelial dysfunction reversal led to decreased mesenteric thrombi in APS mice and may be mediated by HCQ activating extracellular signal-regulated kinase 5 (ERK5) [142,150]. ERK5 has been shown to have endothelial protective effects, including inhibition of leukocyte-endothelial interaction, adhesion molecule expression, and the promotion of laminar flow-induced eNOS expression [151].

Patients are often found to be hypercoagulable following a trauma. Many reasons for this have been identified including the release of platelet-derived extracellular vesicles (PEVs). Although the exact mechanism and function of PEVs is unknown, they are believed to serve initially in promoting hemorrhage control. However, persistent release of PEVs can lead to a pro-thrombotic state and increased thrombin levels. Dyer et al. demonstrated that HCQ is capable of inhibiting the release of PEVs following injury with subsequent decreased thrombus burden in a murine deep vein thrombosis (DVT) model (Figure 4) [152].

## 4. Clinical Trials

Due to the numerous mechanisms of action CQ and HCQ possess, they have been a focus in the development of clinical trials involving multiple disease processes. Table 1 summarizes a number of relevant clinical trials.

### 4.1. Autoimmune Disorders

#### 4.1.1. Systemic Lupus Erythematosus 

Of the diseases treated by CQ and HCQ, systemic lupus erythematosus (SLE) has the most historical data. Despite this, there are still current trials seeking to determine the specific benefits CQ/HCQ has in SLE. In 2005, Fessler et al. observed patients in the multiethnic, observational LUMINA (LUpus in MInorities, NAture versus nurture) study who were within 5 years of SLE diagnosis. Study subjects were followed for several years to observe their disease activity and overall survival. Patients not prescribed HCQ had significantly higher disease activity as measured by the Systemic Lupus Activity Measure (SLAM), along with more accrued damage measured by the Systemic Lupus International Collaborating Clinics Damage Index (SDI) [153]. It was also observed that HCQ had a significant protective effect on patient survival—supporting its continued use by clinicians [154]. Other trials have confirmed improved symptom control with HCQ treatment [155,156].

The impact of CQ on specific organ systems affected by SLE has also been studied extensively. Trials investigating the cardiovascular system have concluded CQ/HCQ reduce the risk of atherosclerosis by lowering total cholesterol and LDL. Additionally, a curative effect on ventricular structure and function following long-term usage has been seen [157,158,159]. Patients with biopsy-proven lupus nephritis who were previously treated with chloroquine have shown a lower frequency of hypertension, infection, thrombotic events, and creatinine >4 mg/dL, compared to those never prescribed the drugs [160]. An et al. took this line of thinking one step further and randomized patients with lupus nephritis to receive either combined immunosuppressive treatment (CIST) with cyclophosphamide, an immunosuppressive agent, and HCQ or to receive cyclophosphamide alone. The CIST group showed both a greater response and complete remission rate [161]. Another potentially life-threatening event of which SLE patients are at higher risk of is thrombus formation. Studies focusing on coagulation demonstrated the protective quality chloroquine-containing medications have against thrombus formation in this specific patient population [162,163].

With SLE being an autoimmune disease, the immune system has been another area of focus with regard to the use of chloroquine. Wozniacka et al. showed the use of CQ to lower circulating proinflammatory cytokines [164]. Evidence exists that macrophage TLR signaling plays a part in maternal anti-SSA/Ro-mediated congenital heart block (CHB). Anti-SSA/Ro is found in several autoimmune conditions, including SLE. Izmirly et al. enrolled mothers who had a previous pregnancy resulting in CHB. They were given HCQ throughout their pregnancy and evaluated for 2nd or 3rd degree heart block. The results showed 4/54 pregnancies were positive for anti-SSA/Ro-mediated CHB—a significant reduction compared to the historic rate for repeated CHB [165].

#### 4.1.2. Rheumatoid Arthritis

Similar to systemic lupus erythematosus, clinicians have utilized chloroquine-derived products for rheumatoid arthritis (RA) for some time, with the specific effects continuing to be researched. In the 1990s, 3 randomized, double-blind controlled trials comparing HCQ to placebo in early or mild RA showed clinical improvements, including decreased corticosteroid injections, as well as improvement in physician and patient assessments of disease progression [166,167,168]. More recently, a placebo-controlled, randomized, multicenter trial measured treatment response after 12 weeks of HCQ treatment using the modified American College of Rheumatology 20 criteria. HCQ patients had a greater improvement compared to their placebo counterparts [169]. Other randomized trials have shown significant improvement in disease treatment when used either in combination with methotrexate, or when used as a single agent [170,171]. With methotrexate remaining one of the most well-known and successful drugs in RA treatment, researchers have sought to further increase treatment options available for patients who fail methotrexate therapy. Multiple trials with this patient population have compared CQ/HCQ to other immunomodulatory agents such as cyclosporin and etanercept, and have shown that while no superiority is seen, all treatment groups have had statistically significant improvement in disease control [172,173,174,175]. These trials support the use of CQ/HCQ in RA patients, particularly those with disease refractory to methotrexate. However, they do not show that it is superior to other forms of treatment.

#### 4.1.3. Antiphospholipid Syndrome

With benefit seen in SLE and RA patients, chloroquine products have been studied for potential use in antiphospholipid syndrome (APS) more recently. For both patients with primary and secondary APS, studies have combined standard of care anticoagulation and/or antiplatelet therapy with HCQ and seen a protective benefit over standard therapy alone [176,177,178]. A 2020 randomized study further showed the addition of HCQ resulted in a lower incidence of thrombosis even after adjustment for other risk factors. Long-term usage of HCQ was also associated with a decrease in APL antibody titers exempting IgM anticardiolipin [176,177,178]. 

Pregnancy in a patient with APS is considered high-risk and can result in miscarriage. A study in 2018 by Ruffatti et al. retrospectively enrolled pregnant patients with primary APS and observed a statistically significant higher live birth rate for patients who were on HCQ compared to other forms of treatment [179]. Currently, there are two trials ongoing, HYPATIA and HYDROSALP, which are attempting to further investigate the benefit this drug may have on protecting pregnancy outcomes [180,181].

### 4.2. Thrombus/DVT Prophylaxis

With the more recent discovery of different medications for thromboprophylaxis, chloroquine-derived products are not often used for this purpose. Nonetheless, there have been studies conducted to see their efficacy in this scenario. Three clinical trials conducted in the 1970s demonstrated significant improvement in DVT prophylaxis with HCQ following major surgery when compared to placebo [182,183,184]. Elsewhere, though, the data in the study by Cooke et al. double-blind, randomized trial failed to show any significant difference in DVTs following elective hip operations [185]. Similarly, Johansson et al. showed platelet aggregability to be inhibited by HCQ in vitro, but failed to see it in vivo [186]. These trials create a mixed review of HCQs standing in thromboprophylaxis. Because of the advancement in other forms of medication therapy for both the prevention and treatment of blood clots, there has been no great interest in continuing to determine the efficacy of the drugs for this specific complication. However, the efficacy of HCQ in cancer—which will be discussed shortly—has resulted in it being considered in cancer patients [187].

### 4.3. Cardiovascular Disease

The primary focus of other clinical trials regarding cardiovascular disease and chloroquine-derived medications revolves around lipids. Three studies involving rheumatic patients found those receiving HCQ had a noticeable benefit in their lipid profiles including a decrease in total cholesterol, low-density lipoproteins (LDL), and apolipoprotein B [188,189,190]. Three additional studies between 2013 and 2015 focused also on chloroquine’s influence on both lipid levels and insulin sensitivity. The consensus of these studies supported a statistically significant benefit regarding lipid panels; however, significance was not always seen in regard to insulin sensitivity [191,192]. Pareek et al. conducted a double-blind, randomized study in 2015 enrolling 328 patients. They found HCQ added to standard atorvastatin therapy resulted in a synergistic effect in the treatment of primary dyslipidemia. Further, they saw significant improvement in patients’ HbA1c and fasting blood glucose—supporting two separate mechanisms for how this drug could reduce cardiovascular disease [193].

The OXI trial enrolled hospitalized patients with recent ST- or non-ST-elevated myocardial infarctions who had undergone diagnostic angiography with or without percutaneous coronary intervention within the previous 96 h. These patients were then randomized to receive six months of either HCQ or placebo and were subsequently followed for 3 years, observing various outcomes and laboratory results [194]. In 2021, an update was published which focused on IL-6—a cytokine attributed to playing a negative role in myocardial infarctions and tissue recovery [195]. Every patient enrolled in the OXI trial had elevated IL-6 at baseline. Those randomized into the HCQ treatment arm showed a significantly lower levels of this cytokine at 6 months compared to the placebo group [196].

### 4.4. Diabetes Mellitus

CQs benefits in diabetes were first observed in patients undergoing CQ treatment for RA. A multicenter observational study of 4905 adult patients with RA noted a lower incidence of lifetime diabetes in those treated with HCQ [197]. Further investigations by other groups have shown HCQ improves insulin sensitivity, beta cell function, and hemoglobin A1c (HbA1c), even in patients refractory to standard treatment [198,199,200,201]. Pareek et al. demonstrated HCQ had the same impact as pioglitazone on glycemic control in uncontrolled diabetics; however, HCQ provided the additional benefit of improving the patient’s lipid panel [199]. Altogether, CQ/HCQ have shown promise in the management of glycemic control in patients with diabetes mellitus.

### 4.5. Cancer

#### 4.5.1. Glioblastoma Multiforme 

Moving beyond the treatment of autoimmune and chronic diseases, there have been endeavors to utilize chloroquine and its derivatives in oncologic research. A small study prescribed long-term CQ alongside standard of care treatment for patients diagnosed with glioblastoma multiforme (GBM). Increased survival time was seen when enrolled patients were compared to their counterparts who received standard of care therapy alone. This led to a conclusion that chronic CQ use may enhance GBM response to antineoplastic treatment [202]. In later trials, safety, efficacy, and treatment benefit were studied. Meaningful results were recorded for achievable autophagy inhibition with HCQ. However, the dose required to achieve the results necessitated dose reduction because of toxicity. Further, neither study showed improvement in overall survival compared to standard of care [203,204].

#### 4.5.2. Pancreatic Ductal Adenocarcinoma 

HCQ was evaluated in 2014 as a potential treatment in cases of metastatic pancreatic ductal adenocarcinoma (PDAC). The response in both humans and murine models was investigated, finding inconsistent autophagy inhibition and insignificant therapeutic efficacy [205]. One year later, Boone et al. conducted a phase 1/2 trial with 35 patients diagnosed with either resectable or borderline resectable disease. Patients were set to receive neoadjuvant gemcitabine in combination with HCQ. The treatment combination was concluded as being safe and well tolerated. Additionally, secondary endpoints of resection rate, overall survival, and percentage increase of autophagy marker LC3-II proved encouraging and supported the continuation of further research [206]. A randomized phase II trial that followed combined gemcitabine/nab-paclitaxel with HCQ and demonstrated significant pathologic and CA 19-9 responses to HCQ treatment compared to those receiving the chemotherapy alone [207,208].

#### 4.5.3. Other Malignancies

CQ has been tested in other oncologic disease processes, though limited data and research has been published [209]. One trial enrolled newly diagnosed breast cancer patients and observed a 15% withdrawal rate due to adverse events related to CQ treatment (including nausea, abdominal cramps, dizziness, visual symptoms, and muscle weakness) [210]. Another trial enrolled patients with multiple myeloma and showed HCQ use alongside immunotherapy was safe and feasible [211]. Altogether, data from CQ use in other oncologic processes are conflicting. Based on HCQs mechanisms and the initial data in oncologic research, future trials and investigation into its cancer application is required.

### 4.6. Viruses

#### 4.6.1. COVID-19

Researchers across the world continue to seek treatments for COVID-19. COVID-19 has many similarities to other viruses with established treatments including HIV, influenza, Ebola, and other variants of coronavirus [212,213]. This includes surface proteins as well as the utilization of cellular endocytic pathways for entry into host cells. Given these similarities, repurposing other anti-viral agents with known safety profiles is an obvious option. Furthermore, advancements in artificial intelligence have allowed rapid evaluation of available data to predict successful agents [214]. CQ received significant media attention from the start of the pandemic. Not only is CQ known to block endocytic pathways, but it also has immunomodulatory effects that had the theoretical ability to control symptoms from virus induced cytokine release [212,213]. In Taiwan, a randomized control trial investigated HCQs impact on duration of disease as measured by PCR testing in hospitalized patients. No difference was seen between the groups [215]. Multiple trials also evaluated HCQs effect on clinical status. For instance, in a multicenter, randomized control trial by Cavalcanti et al., clinical status was evaluated 15 days after diagnosis. HCQ was found to have no additional benefit as compared to standard of care treatment or azithromycin. However, an increased incidence of QT prolongation and elevated hepatic enzymes were noted [216]. A similar study of ICU patients requiring high-flow oxygen, mechanical ventilation, or ECMO again found no significant difference in clinical status at 14 days or 28-day mortality with HCQ compared to placebo [217]. In contrast, a multicenter retrospective observational study of 6493 patients by Arshad et al. demonstrated a significant survival benefit with HCQ treatment. One key difference in these patients was earlier initiation of treatment (median 1 day versus 10–14 days in aforementioned studies) and may explain the many conflicting results in HCQ studies [218,219]. Similarly, Mikami et al. found HCQ was associated with a decreased risk of in-hospital mortality [220].

As the global pandemic continues, efforts to decrease transmission have been pursued. Several trials prescribed HCQ to various groups of individuals—healthy contacts of known patients with PCR-confirmed COVID-19, health care workers at increased risk of developing disease, and household contacts of infected individuals [221,222,223,224,225]. However, no significant difference in transmission could be ascertained, and in some instances, adverse events were noted in those prescribed HCQ compared to placebo. In fact, a number of trials have been stopped early due to safety concerns or lack of sufficient evidence [226,227]. Another randomized trial that utilized HCQ in hospitalized patients was closed early after a lack of sufficient evidence for its efficacy.

#### 4.6.2. HIV

There have been numerous studies showing varying results on the role of CQ and HCQ in the treatment of HIV. HCQ was shown to have strong immunomodulatory activity in HIV patients after a six-month treatment duration demonstrated substantial decreases in IFN-α secreting cells and IL-6, likely due to modulation through the TLR pathway. However, there was no evident increase in CD4+ cells at the end of the treatment period [228]. Conversely, a subsequent equally powered study found no overall decrease in HIV-induced inflammation when a lower dose of CQ was prescribed for the same duration [229]. Importantly, both of these studies were in patients receiving concurrent HAART therapy. In a double-blind, randomized, placebo-controlled trial, there was an observed decrease in CD4+ cells, an increase in viral load, and a significant increase in flu-like symptoms when HCQ was administered to HIV-positive patients not receiving HAART therapy [230]. Another study showed no significant effects on modifying immune activation in HIV and noted a substantial increase in infection rate of astrocytes [231].

#### 4.6.3. Flaviviruses

Zika virus is a flavivirus known to cause congenital microcephaly, likely secondary to loss of neural progenitor cells (NPC) following infection. Chloroquine, as described, has a large Vd with CSF levels up to 30 times higher than plasma [232]. This is a proposed reason why chloroquine was shown to be effective in reducing Zika infection in human fetal NPC and mouse models [233]. Efficacy of chloroquine in prevention of microcephaly was seen to be greater when administered earlier in the infection period, likely due to greater conservation of NPCs due to decreased viral entry by inhibition of endocytosis [233,234,235]. Chloroquine has also been studied when used to treat another flavivirus, hepatitis C. Following treatment, HCV positive patients have shown regression in porphyria cutanea tarda skin lesions (effects known to be associated with successful HCV treatment) [236]. Ferroquine, an analogue of chloroquine, has also been shown to be a strong inhibitor of HCV entry and replication in hepatic cells [237].

**Table 1 pharmaceutics-14-02551-t001:** Review of clinical trials evaluating CQ and HCQ in human disease.

Author (Year) [Ref]	Design	Intervention	Outcome
**Systemic Lupus Erythematous (SLE)**			
Fessler et al. (2005) [153]	Observational	HCQ	Lower disease activity and improved survival in patients receiving HCQ
Vilela et al. (2001) [238]	Phase II RCT	Prednisone and either HCQ or placebo	Lower disease activity and reduction in prednisone dose in HCQ group. No adverse side effects in infants of mothers receiving HCQ
Canadian Hydroxychloroquine Study Group. (1991) [156]	Phase 3 RCT	Clinically stable SLE patients receiving HCQ for at least 6 months randomized to continue HCQ treatment or placebo	2.5 × higher relative risk of clincal flare-up in placebo group
Ruiz-Irastorza et al. (2006) [162]	Observational prospective cohort study	CQ or HCQ treatment versus non-antimalarial treatment	Decreased thrombosis incidence and increased survival rate
An et al. (2019) [161]	Phase III RCT	Immunosuppressive treatment alone or in combination with HCQ for lupus nephritis	Higher rate of complete remission in combination group
**Rheumatoid Arthritis**			
Das et al. (2007) [169]	Phase III RCT	Nimesulide 100 mg BID plus either 200 mg HCQ daily or placebo for 8 weeks	Symptomatic improvement with HCQ based on ACR 20 criteria
Gubar et al. (2008) [171]	Prospective randomized study	MTX (17.5 mg/week) alone versus in combination with SSZ (2.0 g/day) and HCQ (200 mg/day) for 1.5 years	Triple combination therapy with MTX, SSZ, and HCQ had improved symptomatic response (ACR > 50%)
O’Dell et al. (2013) [174]	Phase III RCT	Patients with active disease despite MTX randomized to either triple therapy (MTX, SSZ, HCQ) or Etanercept plus MTX	Significant improvement in disease activity score in both groups. Triple therapy was noninferior to Etanercept plus MTX
**Antiphospholipid Syndrome**			
Kravvariti et al. (2020) [176]	Phase III RCT	HCQ plus standard anticoagulation and/or antiplatelet therapy versus standard care alone	Lower incidence of thrombosis with HCQ plus standard care than with standard care alone
Schmidt-Tanguy et al. (2013) [177]	Prospective non-randomized study	HCQ versus standard oral anticoagulants	No difference in thrombosis incidence
**Cardiovascular Disease**			
Pareek et al. (2015) [193]	Phase III RCT	Atorvastatin alone versus in addition to HCQ in patients with primary dyslipidemia	Significant reduction in LDL in Atorvastatin/HCQ patients but no change in triglycerides or HDL.
Ulander et al. (2021) [196]	Phase II RCT	HCQ versus placebo after myocardial infarction	Lower IL-6 levels with HCQ treatment without higher adverse reactions
**Cancer**			
Rosenfeld (2014) [204]	Phase I/II trial	HCQ with radiation and adjuvant temozolomide in glioblastoma multiforme patients.	Thrombocytopenia and Grade 3 and 4 neutropenia at 800 mg per day HCQ. Maximum tolerated dose was 600 mg per day with radiation and temozolomide.
Zeh et al. (2020) [207]	Phase II RCT	Two cycles of nab-paclitaxel and gemcitabine alone or with hydroxychloroquine in patients with potentially resectable pancreatic cancer	Improved histopathologic and CA 19-9 responses with addition of HCQ. No difference in severe adverse reactions.
Karasic et al. (2019) [208]	Phase II RCT	Nab-paclitaxel alone or with 600 mg HCQ BID in patients with previously untreated metastatic pancreatic cancer	No difference in overall survival at 12 months
Arnaout (2019) [210]	Phae II RCT	HCQ 500 mg daily versus placebo in newly diagnosed breast cancer patients	No change in cellular proliferation. Although all adverse effects were classified as grade 1, 15% of patients receiving HCQ withdrew from the study.
**Viral**			
Cavalcanti et al. (2020) [216]	Phase III RCT	Standard care alone or in combination with HCQ +/− Azithromycin in patients with mild to moderate COVID-19 patients	No improvement in clinical status at 15 days as compared to standard care
Abella (2020) [222]	Phase II RCT	HCQ versus placebo as COVID-19 prophylaxis in healthcare workers	No difference in infection rates

RCT = Randomized Controlled Trial, BID = two times a day, ACR = American College of Rheumatology, MTX = Methotrexate, SSZ = Sulfasalazine, aPL = Antiphospholipid Syndrome, IL = interleukin, LDL = low-density lipoprotein, HDL—high-density lipoprotein.

## 5. Discussion

As seen, chloroquine and hydroxychloroquine have proven beneficial for many disease processes throughout the past century. Over time, enhanced understanding of the many mechanisms of action for these drugs has increased exponentially and opened doors to its utilization in other diseases and conditions. We identified five major areas of CQ/HCQ function in our review including alkalinization of lysosomes and endosomes, downregulation of CXCR4 expression, HMGB1 inhibition, alteration of intracellular calcium, and prevention of thrombus formation. As illustrated in the preceding figures, these areas contain multiple overlapping pathways and highlight that prior research efforts may be focused on points past the site of pathway convergence. Although the effects of CQ/HCQ are likely multifactorial, future research should investigate upstream components of the proposed mechanisms of action in order to truly understand the workings of these agents. While CQ/HCQ’s clinical effects have been shown in several pathologic processes such as SLE, RA, and as an antimalarial, further understanding of its various roles has the potential for treatment of disease. In particular, future studies in the field of oncology prove promising to show a benefit of this drug in the ongoing fight against cancer.

## Figures and Tables

**Figure 1 pharmaceutics-14-02551-f001:**
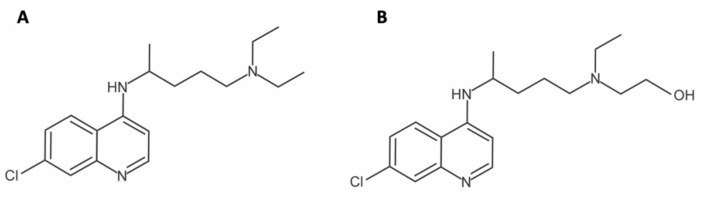
(**A**) Structure of Chloroquine (**B**) Structure of Hydroxychloroquine.

**Figure 2 pharmaceutics-14-02551-f002:**
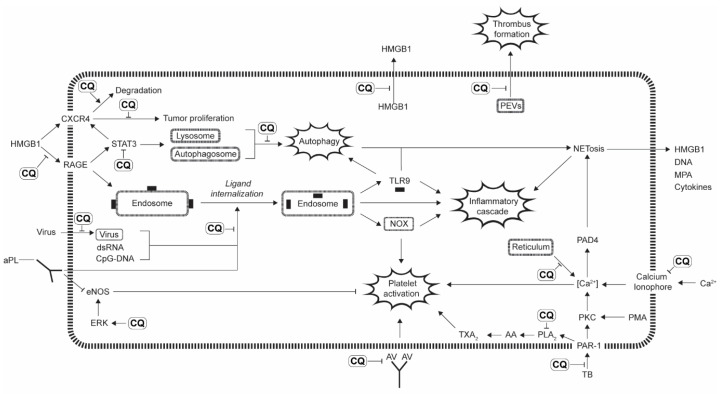
Summary of proposed mechanisms of actions of CQ.

**Figure 3 pharmaceutics-14-02551-f003:**
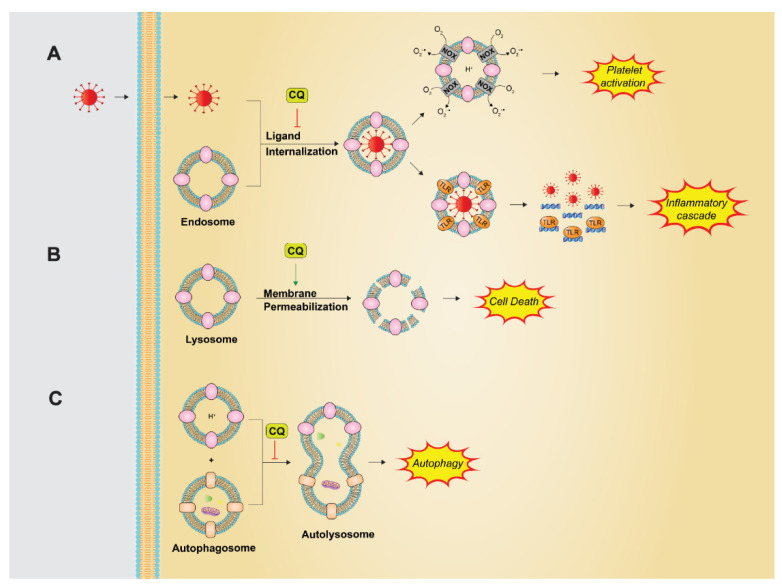
The impact of chloroquine and hydroxychloroquine on endosomal and lysosomal function. (**A**) Endosomal internalization of ligands allows for cell signaling. This process requires an acidic environment which CQ inhibits, preventing downstream signaling through endocytic TLRs and the NOX complex. (**B**) CQ induces lysosomal membrane permeabilization, leading to cell death. (**C**) CQ prevents lysosomes from merging with autophagosomes in the process of autophagy through lysosomal alkalinization.

**Figure 4 pharmaceutics-14-02551-f004:**
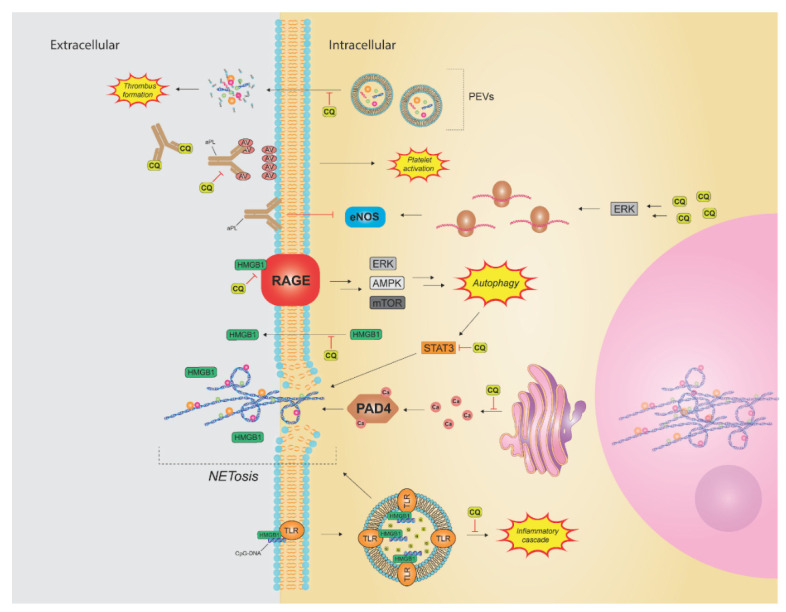
Inhibition of thrombus formation and NETosis. Thrombus formation driven by platelet-derived extracellular vesicles (PEVs) and antibody-induced platelet activation is inhibited by CQ. CQ also has a complex interference with NETosis, shown in the figure via inhibition of RAGE-mediated autophagy, HMGB1 release, and activation of PAD4, NOX, and TLRs.

**Figure 5 pharmaceutics-14-02551-f005:**
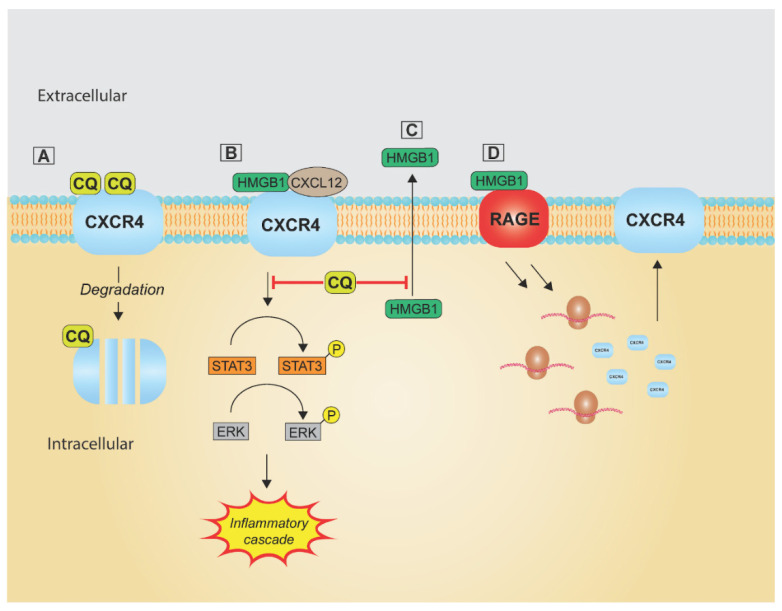
(**A**) CQ upregulates C-X-C Chemokine Receptor Type 4 (CXCR4) degradation. (**B**) CQ impedes CXCR4 signaling following HMGB1 activation. (**C**) CQ prevents HMGB1 release. (**D**) CQ hinders RAGE-mediated CXCR4 upregulation.

**Figure 6 pharmaceutics-14-02551-f006:**
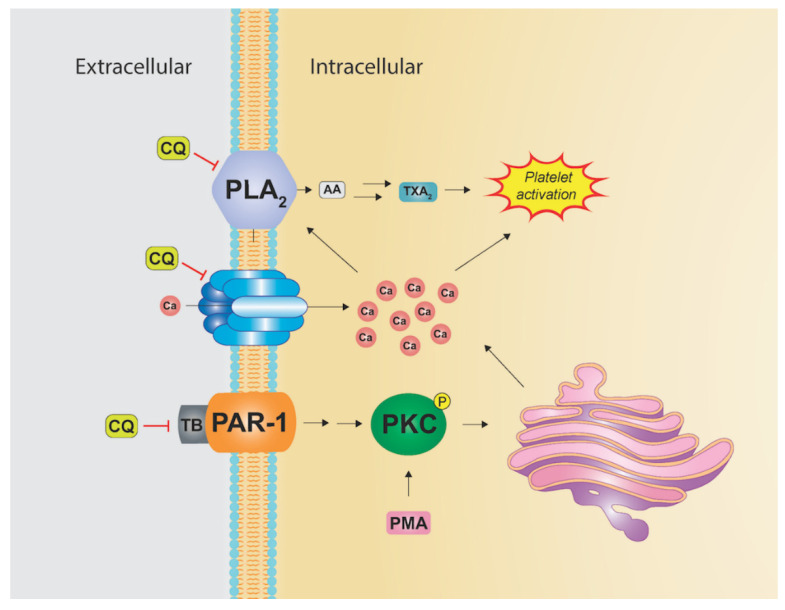
The influence of CQ on intracellular calcium concentration. CQ inhibits increases in intracellular calcium via calcium ionophores, thrombin (TB), and phorbol-myristate-acetate (PMA). Elevated intracellular calcium can stimulate release of phospholipase A2 (PLA2), leading to arachidonic acid (AA) liberation and subsequent platelet activation.

## Data Availability

All data generated or analyzed related to this publication are included in the published article.

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
