# Peer review of "Biologic Functions of Hydroxychloroquine in Disease: From COVID-19 to Cancer"

_pharmaceutics, 2022, doi:10.3390/pharmaceutics14122551_

Round 1

Reviewer 1 Report

This manuscript reviews the mechanism of action and clinical data of QC and HQC for multiple diseases treatment. Firstly, the authors briefly introduce the drug discovery history of QC/HQC as anti-malarial drugs and the repurposing of them for autoimmune diseases, viruses, and cancers, etc. In addition, the pharmacokinetics and side effects of QC/HQC are mentioned that they possess high bioavailability with F values from 78%-100% in different dosage forms associated with good distribution properties. However, long term treatment with them at a dosage of 10-15mg/kg has caused a series of side effects, such as retinopathy, puritus, and insomnia. Then alkalinization of lysosomes and endosomes, downregulation of CXCR4 expression, HMGB1 inhibition, alteration of intracellular calcium, and prevention of thrombus formation are summarized as QC/HQC biologic functions. Finally, the current statuses of clinical trials of QC/HQC for autoimmune disorders, thrombus/DVT prophylaxis, cardiovascular disease, diabetes mellitus, cancer, and virus are described in detail, respectively. In general, this review is impressive and contains substantial content. However, there are some issues/problems need to be addressed before ready for acceptance. For example, “clinical trials of QC/HQC are not sufficiently linked to “mechanism of action”, and furthermore, the conclusions of this review and the authors' prospect based on the current state of QC/HQC research are not sufficiently elaborated. Therefore, I suggest major revision of this review, some of the concerns included below.

1)      Authors must add references to the description, “Chloroquine (CQ) and its derivative, hydroxychloroquine (HCQ), are well-known, multi-use drugs with applications in anti-malarial and anti-viral treatment, autoimmune diseases, and neoplastic processes”, at the beginning of the introduction section.

2)      Corresponding structures of the main compounds mentioned should be displayed in the review.

3)      Before review the repurposing of QC/HQC, the main reasons for chloroquine resistance in the treatment of malaria at the molecular level should be discussed.

4)      Like other mechanisms of action of QC/HQC, a schematic representation of the mechanism by which HQC prevents thrombosis should be supplemented.

5)      The authors mentioned the clinical trials of QC/HQC in the treatment of autoimmune disorders, cardiovascular disease, cancer and viral infection, and what is their relationship with the aforementioned mechanism of action of QC/HQC.

6)      On page 15. Why the viral load increases when HCQ was administered to HIV-positive patients not receiving HAART therapy.

7)      In order to more intuitively understand the clinical research status of QC/HQC, the authors should provide a corresponding table to show the types of diseases treated by QC/HQC and the specific detailed clinical trial stages, etc.

8)      The content of the discussion section is too simplistic, and it is difficult to know what the author's conclusions are after reviewing the mechanism of action and research status of QC/HQC, what are the relevant issues need be urgently solved at present, and what are the author's insights.

Reviewer 2 Report

There are many reviews on this molecule very recently. The authors should explain what new this review adds if anything. A review on a redundant topic is hard to find readers. Why authors choose to not start with the CQ effect on Plasmodium growth and how it has been used as a tool to study parasite biology is beyond my understanding. Reading the review it feels to make it different they just avoid its main usage and history behind the chloroquine resistance,

This nearly century-

old drug has an additional benefit of interest to researchers and clinicians, in that it is 57

extremely affordable with tablets costing approximately $0.01-0.02 each and solutions 58

costing $0.01-0.02 per milliliter for 10mg/ml concentration [21]. Cost to any drug depends on many factors. I am not sure here if authors mean production cost or cost in which it is available to the public. In either case I suggest just suggesting it is available cheaply without going into numbers.

The only contraindications for drug administration are known hypersensitivity to 4- 110

aminoquinoline compounds, or treatment of any diseases—other than acute malaria—in

the presence of retinal or visual field changes of any etiology Sentences need revision. I am sure what the authors mean here.

Figure 1 needs improvement.

LMP induction may be an additional mech- 208

anism by which CQ overcomes resistance to chemotherapy What is LMP ?

For example, Cisplatin, a 209

DNA crosslinking chemotherapy, is used in non-small-cell lung cancer (NSCLC) pa- 210

tients who are not eligible for immunotherapy; however, resistance forms quickly. Sentence is not clear.

Additionally, CQ and HCQ treat yeast infection via iron deprivation, 229

and this has also been proposed as a mechanism of its anti-viral potential [85-87]. Not sure what Cq has to do with iron deprivation. This sentence is unclear and only adds confusion.

Treatment with CQ decreased AKI, TLR protein in the spleen, systemic inflammation, 239

and improved mortality [95]. This does not make sense. How CQ decreases toxicity when it is known to have toxicity itself. Further what authors mean by improved mortality ? I suspect the authors do not understand the original study and have done a poor job in this sentence.

Further revealed was CQ’s capability to directly bind TLR ligands, pre- 248

venting their binding to receptors. Improve the sentence.

Figure legends are missing.

Reviewer 3 Report

The review is sound and comprehensive though it is addressing an issue that is extensively investigated.

Round 2

Reviewer 1 Report

I am happy with the revisions made by the authors and suggest acceptance of the manuscript in its current form.

Author Response

Thank you for reviewing our manuscript and for your support of its publication. 

Reviewer 2 Report

1.      Abstract is too small and vague. It needs to provide some information about specific information of the topic and what this review focusses on

2.      For any review it is important that referencing is correct. In this review I find many issues with referencing which suggests poor knowledge of the subject. I will cite one example here “Through autophagy inhibition, CQ and HCQ prevent autoantigen presentation in antigen presenting cells and B cells, resulting in decreased T cell activation and overall blunted activation” The reference cited here is a review by Xourgia et al. This review does not discuss role of chloroquine on T-cell activation. Further vague information like “overall blunted activation” indicates authors are not taking effort in explaining this properly. I can cite many more examples but my point is before final decision authors needs to go through the reference list and ensure the information is correct.

3.      Although the effects of CQ/HCQ are likely multifactorial, future research should investigate upstream components in or- der to truly understand the workings of these agents.” I am not sure what authors means by upstream component here.

4.      I suggest to include some background on COVID-19 and highlight the fact that drug discovery remains a challenge (PMID: 34451513 , 34055887 , 32325767 ).
